# Reasons for Low Protection of Vulnerable Workers from COVID-19—Results from the Quantitative and Qualitative Study on Working Life in Latvia

**DOI:** 10.3390/ijerph18105188

**Published:** 2021-05-13

**Authors:** Linda Matisāne, Linda Paegle, Maija Eglīte, Lāsma Akūlova, Asnate Anna Linde, Ivars Vanadziņš, Iveta Mietule, Jeļena Lonska, Lienīte Litavniece, Iluta Arbidāne, Sarmīte Rozentāle, Ieva Grīntāle

**Affiliations:** 1Institute of Occupational Safety and Environmental Health, Rīga Stradiņš University, Dzirciema 16, LV-1007 Riga, Latvia; Linda.Paegle@rsu.lv (L.P.); Maija.Eglite@rsu.lv (M.E.); Lasma.Akulova@rsu.lv (L.A.); AsnateAnna.Linde@rsu.lv (A.A.L.); Ivars.vanadzins@rsu.lv (I.V.); 2Research Institute for Business and Social Processes, Rēzekne Academy of Technologies, Atbrivosanas Alley 115, LV-4601 Rezekne, Latvia; Iveta.Mietule@rta.lv (I.M.); Jelena.Lonska@rta.lv (J.L.); Lienite.Litavniece@rta.lv (L.L.); iluta.arbidane@rta.lv (I.A.); 3Institute of Social, Economic and Humanities Research, Vidzeme University of Applied Sciences, Cesu Street 4, LV-4201 Valmiera, Latvia; Sarmite.Rozentale@va.lv (S.R.); Ieva.Grintale@va.lv (I.G.)

**Keywords:** COVID-19, vulnerable workers, occupational health, workplace risk assessment, chronic diseases

## Abstract

Several individual factors like older age and chronic diseases have been linked with more severe symptoms often leading to hospitalization and higher mortality from COVID-19. Part of adults with such factors is still active in the workforce. The objective of the study was to identify measures taken by the employer to protect them and to investigate reasons for low protection of vulnerable workers during the 1st wave of the COVID-19 pandemic. Answers from 1000 workers collected via web-survey and results from 10 focus group discussions were analyzed. Only 31.5% of respondents mentioned that their employer had identified existing vulnerable groups and offered specific measures to protect them. Moving vulnerable workers away from the workplace was the most frequent measure (e.g., transfer to the back-office without contact with clients, telework, paid vacations, paid downtime). Most employers do not see elderly workers and workers with chronic diseases as risk groups, thus are not specifically protecting them. Instead, several employers have included workers critical for business continuity in their risk group. Others had not taken measures because of the lack of information due to general data protection regulation. Poor communication and lack of interest of employers to ask their workers if they need special protection is the topic to be addressed at the national level.

## 1. Introduction

As a response to the global spread of COVID-19 and depending on the local situation, governments have taken different measures to reduce the spreading of COVID-19, including school and workplace closures, cancellations of public events, stay-at-home advise, self-isolation, restrictions on internal and external movements, etc. One of the reasons for those decisions is to protect national healthcare systems from overloading [1]. Another reason is to prevent vulnerable persons (people with elevated risk) from severe progress of COVID-19 infection and other poor outcomes which can result either from individual factors or workplace factors [2].

Older age, being a man, belonging to non-white ethnicity, smoking, having low income and suffering from chronic health conditions, such as overweight and obesity, diabetes, hypertension and other cardiovascular diseases, respiratory diseases, kidney diseases, compromised immunity (including cancer and autoimmune diseases) have been linked with more severe COVID-19 symptoms often leading to the development of acute respiratory distress syndrome, risk of hospitalization, admission to intensive care unit, intubation and higher mortality from COVID-19 [3,4,5,6,7,8]. In addition, lower education level is indirectly associated with increased risk of developing severe forms of COVID-19 through different mechanisms, such as increased prevalence of smoking and poor nutrition, which could suppress the immune system [3].

Older adults are often having chronic health problems, and the risk increases with age [9]. Thus, in the UK 70% of all deaths related to COVID-19 are in the age group over-70-years, and nearly two-thirds (64%) of the remaining deaths have been reported in the 60–69 years age group [10]. In China, the case fatality rate was less than 0.5% among people under 50 years of age, 1.3% among those 50 to 59, and 3.6% among those 60 to 69 [4]. However, a metanalysis of five studies showed that increased age-related risk of COVID-19 disease severity, admission to intensive care units, and death is not an isolated effect of age. If the important age-related risk factors (like diabetes, hypertension, coronary heart disease/cerebrovascular disease, compromised immunity, previous respiratory disease, renal diseases) are taken into account, then there is a slightly (2.7%) increased risk per age year for COVID-19 diseases severity and almost no age-related risk for death [6]. Previous experience from frail patients suggests that identification of conditions of susceptibility requiring specific preventive actions to possibly avoid the infection is important to achieve better management of the disease [8]. This means that individual risk for all workers should be assessed in order to define low-, medium-, and high-risk categories of death from COVID-19; and these categories are based on age and the presence of high-risk chronic conditions (where low risk means younger age without high-risk condition; medium risk means middle age or younger age with a high-risk condition; high risk means older age or middle age with the high-risk condition) [4].

Part of those adults belonging to vulnerable groups is still active in the workforce in Latvia as in Latvia retirement age from 1 January, 2021 is 64 years [11]. The results of the survey “Health Behavior among Latvian Adult Population, 2018” show that a high proportion of the study population in the age group 55–64 have self-reported those diseases which are associated with more severe COVID-19 cases. Hypertension has been reported by 32.9% male and 47.1% female respondents, and most probably the actual prevalence of hypertension is even higher due to the fact that within the previous year blood pressure has been measured only for 78.0% of male respondents of the relevant age group and 92.8% in the relevant female group. Diabetes has been mentioned by 7.6% male and 9.4% female persons in the age group 55–64. A total of 10.0% of interviewed males and 13.3% of females in the same group have pointed out having heart failure. During the same study calculated body mass index based on measured anthropometric measurements was determined. Overweight was reported for 41.6% of males aged 55–64 and 34.0% of females in the same group [12]. Results of another survey show that in Latvia in September 2020, 23.7% of inhabitants believed they belonged to vulnerable groups concerning COVID-19 with severe consequences, the prevalence of such persons was higher among older workers (above 60%) and persons with average income (above 30%) [13]. This means that every single company employing older workers has COVID-19 vulnerable persons and always has to consider special individualized preventive measures to protect them in the workplace.

The risk of becoming COVID-19 positive with contacts at work mainly depends on the prevalence of COVID-19 in the local community [5]. Staying at home has shown the best results to reduce infection rates, however, there are still a rather large proportion of workplaces that cannot be adjusted to distance work [14]. In such cases, occupation and/or nature of work may expose workers to SARS-CoV-2 during the work process. In situations when interfacing with the public, being in close distance with other workers, interaction with other persons (e.g., clients), or caring for people is an integrated part of the job, COVID-19 can be transmitted both ways—from the public to the worker and from the worker to the public [2,3]. If assessing the type of work, low risk for COVID-19 transmission means no contact with COVID-19 positive persons (e.g., work from home), medium risk means contact with people with uncertain COVID-19 status, but high risk means contact with people known to be COVID-19 positive [4]. Persons preparing food, health and social care workers, sales and retail staff, cleaning staff, teachers, the crew onboard cruise ships as well as migrants and low skilled occupations are at higher risk [2,3,15,16]. Due to the challenges with the supply of proper personal protective equipment during the 1st wave of the COVID-19 pandemic, health care workers were at increased risk to fall ill themselves and to spread the virus further [17]. Having no previous experience and suddenly facing a new lethal virus, employers were not ready to implement their legal obligations to sufficiently protect their workers in the workplace [18]. Partly this was related to the fact that orders issued by national governments did not contain any specific guidance for employers on how to protect vulnerable workers who are at increased risk for poor outcome of COVID-19 because of advanced age or chronic conditions. Sectors and companies having previous experience with the protection of workers from biological agents (e.g., food processing, health care) managed better as they already had awareness, infectious disease prevention plans, and policies in place [2]. Appropriate COVID-19 response plans at the company level would include workforce planning, integrating infectious disease and epidemiological safety training into other workplace training, patient streaming by the risk of infection, mask-wearing, and infection control precautions, including use of personal protective equipment, etc. [2,19]. In addition, management of COVID-19 is not simple—from one side the employers need to understand and grade individual risk factors for all workers, from the other–these individual conditions must be interpreted concerning the possibility of exposure to COVID-19 at work [10,12].

Most of the companies were not ready for COVID-19, they needed support in the adoption of suitable preventive and protective measures from occupational health and safety (OSH) experts, medical professionals, physicians, and professional organizations, including advice for individual risk assessment which leads to the need of individually adapted and tailored preventive and protective measures [2,4,8].

During individual risk assessment, two main aspects should be taken into account: individual parameters of worker (chronic health conditions, age, personal habits) and type of work (contact with known or possibly COVID-19 positive persons). For persons with high risk in both domains, it is advisable to stop their work temporarily, and those with high risk in one domain and medium risk in the other should discuss risk with their physician [4]. Many workers are not able to stop working without additional financial support either provided by the employer or government which means that special support solutions are needed to protect vulnerable workers in high-risk workplaces (e.g., increased access to paid sick leave, hazard pay for those exposed during a pandemic, etc.) [2,4]. It should be specified that no financial support from the state for selected vulnerable groups participating in the labor market was available in Latvia during the first emergency state.

However, there are some industries like petrochemical plants, power plants, water treatment plants, food production companies, and others, which cannot stop the work even during the pandemic–workers should work onsite in their workplaces to provide products/services [20]. If the workers are not able to stop working, then measures at workplaces should be taken. It is advisable to use a hierarchy of controls: (a) removal or reduction of exposure at the source (flexible work solutions, where possible, screening, testing, case investigation, contact tracing, etc.); (b) redesigning of the work environments to facilitate social distancing (e.g., installation of barriers, other protection elements), promotion of frequent hand washing and sanitizations; (c) adoption of organizational preventive measures in the workplaces; (d) promotion of OSH and epidemiological education to workers and other measures to minimize the exposure (e.g., use of personal protective equipment) [8]. 

Previous research has already pointed out the need for more knowledge on COVID-19 risk assessment for different groups of essential and non-essential workers and management for workplace risk factors and workers, particularly for vulnerable individuals [8,16].

The objective of our study is to identify preventive measures provided by the employers for the vulnerable groups (elderly workers and workers with chronic health conditions) and to assess the possible reasons for the low protection of those groups during the 1st emergency state in Latvia which in force between 12 March and 9 June, 2020 due to the COVID-19 pandemic. It provides additional data for better protection of vulnerable workers from the risks of COVID-19 in the workplaces. This research is original and provides evidence on the need for more active implementation of preventive measures at the company level to protect vulnerable groups.

## 2. Materials and Methods

Quantitative and qualitative research methods were used to gather information: a web-based questionnaire was used to survey workers in Latvia and focus group discussions were organized to gather information from employers and their representatives—OSH experts. Ethical approval for the study was granted by the Ethics Commission of Rīga Stradiņš University (protocol No. 6-1/08/16, 23 July 2020) before the recruitment of any participants.

### 2.1. Web-Survey of Workers

#### 2.1.1. Recruitment and Data Collection

Web-survey as an online tool was used to quickly gather information from workers between 28 September and 27 October 2020 on measures taken by their employers to mitigate the spread of COVID-19 in their workplaces. It applied a non-probability sampling method. Survey participants were recruited using a snowball sampling method, social media advertisements as well as direct emails to share the web link of the questionnaire in Latvian. Every single person having access to the internet was able to fill in the questionnaire. The same recruiting principle was used during the Eurofound survey “Living, working and COVID-19” [21]. Survey data were gathered and managed using REDCap (Research Electronic Data Capture) tool.

At the beginning of the web-survey, filtering questions were applied to recruit only paid workers who were employed during the previous year. The following exclusion criteria were used: working without salary in family businesses, working without salary on family farm, being on maternity leave, being unemployed persons, being only retired persons, being housewives, being only school-children or students during the survey period.

While designing the survey, the survey sample size was calculated, using 5% margin error, 99% confidence intervals, 50% response rate, and 892,100 employed persons in Latvia in the 2nd quarter of 2020 [22], resulting in 663 persons. To increase the probability of finding statistically significant results and taking into account the planned time frame of the survey, the authors decided to make the web-link available one full calendar month or until the moment when there will be 1000 fully-filled answers, whichever will occur first. In this case, the link to the web-survey was locked on the next morning of workday after 1000 respondents have answered all of the survey questions.

In total, 1823 persons responded to the questions, however, only 1006 respondents answered all questions (response rate—55.2%). Data on the number of clicks on the web-survey landing page is not available. To be able to apply weights, additional 6 persons were excluded from the analysis due to that lack of information (e.g., not willing to specify gender or age).

For the needs of this article, two additional groups of respondents were excluded from further analysis as they were not working during the first emergency state and, therefore, they were not able to report preventive measures taken by the employer. These groups included persons who lost their jobs and did not find new employment during the 1st emergency state and persons who were on state paid downtime the whole emergency state period. Therefore, 878 respondents were included in further analysis. The average age of these respondents was 43.8 (SD = 11.0, min 19, max 72 years), 21.1% were males and 78.9% females. A detailed description of the study sample is available in Appendix A (Table A1). At the beginning of the web-survey, written information on the purpose of the study was provided, therefore, participants by voluntary proceeding to the questions agreed to participate in the survey.

#### 2.1.2. Study Variables

Respondents were asked to give feedback on seventeen different statements regarding measures that can be implemented in the workplaces to reduce the spreading of the COVID-19 virus among workers (all statements are given in Appendix A, Table A2). The statements to be included in the questionnaire were selected to cover national legal requirements (e.g., on stay-at-home policy, use of disinfectants) in force during the 1st emergency state caused by the COVID-19 and measures provided by the employers in Latvia which had been identified as good practice examples by the State Labour Inspectorate and the Institute for Corporate Sustainability and Responsibility and published on the national occupational health and safety website www.stradavesels.lv on 8 May 2020 [23]. The questionnaire was initially drafted by three authors (L.M., J.L., S.R.), then evaluated by other two experts (I.V., I.A.), and tested by three other experts (L.P., I.A., I.G.). Based on the received comments, the instrument was improved and sent for review to the Ministry of Welfare who was the main stakeholder to use the obtained results from the project “Life with COVID-19: Evaluation of Overcoming the Coronavirus Crisis in Latvia and Recommendations for Societal Resilience in the Future” (VPP-COVID-2020/1-0013). After this approval, the questionnaire was programmed and tested for readability, consistency of style, formatting, and the clarity of the language by five independent persons who were not involved in the study and have no background related to occupational health and safety.

A set of filters and rooting was applied to the questionnaire to adapt the questionnaire to the particular situation of the respondents, e.g., the telework section was asked only to those who reported their work can be done from distance and they worked from home during the first emergency state due to the COVID-19 pandemic.

For this article, the answer to the following statement was used “Workers belonging to risk groups were identified and special work conditions were offered to them (workers with chronic diseases, workers older than 55 years)”. Several answers were possible: “It was necessary and was provided in all cases”, “It was necessary, but was provided only in some cases”, “It was necessary, but was not provided”, “It was not necessary and was not provided” (in the article referred as “provided”, “partly provided”, “not provided”, “not needed”), “I don’t know/hard to say”.

#### 2.1.3. Statistical Analysis

Descriptive analyses (mean, standard deviation) and frequency analyses (percentages, distribution) were used to describe the data. To obtain data that is representative of the demographic profile of the working population in Latvia, the sample was weighted based on gender and age. Data weights were made by age crossed with gender (in 12 age-gender combinations) and data were analyzed with statistical software IBM SPSS, version 26 (IBM Corporation, Armonk, New York, NY, USA). Weighting targets included population estimates of the 3rd quarter of 2020 by age groups and gender obtained from the Central Statistical Bureau of Latvia.

Results with applied weights are described in the section “Results”, results without applied weights and with applied weights are given in Appendix A, Table A2.

### 2.2. Focus Group Discussions

#### 2.2.1. Study Design and Recruitment

Focus group discussions as a qualitative research method are becoming more and more popular [24,25]. To obtain qualitative data on measures taken by employers at the company level to mitigate the spread of the SARS-CoV-2 virus, including protection of vulnerable workers in total 10 focus group discussions were held.

Focus group participants were recruited on a voluntary basis through public announcements, social media (Facebook and Twitter) posts, local employers’ non-governmental organizations, personal contact networks, and national labor inspectorate. Before the discussion, participants were fully informed about the purpose of the study, and therefore, verbal consent was obtained. Participants did not receive any monetary compensation for their participation.

#### 2.2.2. Focus Group Discussions of Employers

During 8 focus group discussions (with at least 5 and a maximum of 11 participants) 65 employers from companies of different sizes and regions were interviewed to collect adequate contributions for comparisons between different groups of employers depending on the number of workers in the organization and region where the organization is located. Taking into account the structure of economics in Latvia, for this study, organizations with a number of workers below 100 were classified in the group of small and medium-sized organizations, but companies with 100 or more workers were classified as large. During the recruitment process, the affiliation of the possible participant was checked to ensure that the person can represent the employers’ opinion. If the person did not meet this requirement, he/she was excluded from participation in the focus group. If the person applied to participate in the focus group discussion, but he/she did not match the requirements for the specific group (e.g., size of the company or geographical location of the company), he/she was offered to participate in the relevant group or excluded from participation. Detailed characteristics of the focus group participants for employers are available in Appendix A (Table A3).

#### 2.2.3. Focus Group Discussions of OSH Experts

Two focus group discussions with 12 participants per group were held with OSH experts (one group focusing on the experts working as external OSH service providers, another one—for company internal OSH experts). During the recruitment process the status of the OSH expert was checked through their affiliation (in the case of internal OSH experts) or in the official public list of registered OSH service providers). If the person applied to participate in the focus group discussion, but he/she did not match the requirements for the specific group (e.g., internal/external OSH expert), he/she was offered to participate in the other group or excluded from participation. The only exceptions were representatives from the employers’ non-governmental organization and trade unions who could provide experience from their member organizations. Detailed characteristics of the focus group participants for OSH experts are available in Appendix A (Table A4).

#### 2.2.4. Procedure of the Focus Group Discussions

All focus group discussions followed a standardized procedure. Due to COVID-19 pandemic restrictions, a mixed interviewing method was used—part of the group was interviewed on-site, others were interviewed using online platforms—Zoom and MS Teams. Focus group discussions were led by experienced and trained moderators (interviewers—I.V., I.A., S.R.) and facilitated by a note-taker according to structured guidelines (research protocols) with logically proceeding questions. Part of the guidelines was identical for employers and OSH experts, but the other part differed (e.g., on very narrow and specific topics related to OSH performance). Pre-testing of guidelines was organized with persons familiar with OSH-related topics, but who were not involved in the research.

The topic related to the protection of vulnerable groups was included towards the end of the group discussions when general topics applying to all workers were already covered. According to the structured guidelines, participants were initially asked if the companies in any way had identified workers who might belong to increased risk groups concerning COVID-19, but these groups were not defined by the moderator from the very beginning. After spontaneous answers further as open-ended as possible questions focused on senior workers and workers with chronic diseases. The discussion flow was then directed to cover implementation of special measures like encouraging telework, the offer of being off workplace, an increase of disinfection in the workplace of vulnerable workers, use of additional distancing measures in the workplace of vulnerable workers, additional training on preventive measures, etc. This section finished with a discussion on the behavior of vulnerable workers—if they were enthusiastic and accepted the offer of the employer.

As part of the discussions were held online, a PowerPoint presentation was used to share the questions when the relevant questions were discussed. All the focus group discussions were recorded with the permission of participants to facilitate the transcribing process and to ensure that the information is matched correctly. Recordings are safely stored according to the data protection rules of Riga Stradiņš University. The length of group discussions was between 115 and 152 min each, topics related to vulnerable groups were covered in approximately 10 min per focus group.

#### 2.2.5. Data Analysis

After focus group discussions anonymized transcripts were prepared and the participants were de-identified manually. To maintain the anonymity of the participants, de-identification was carried out by an independent researcher (L.A.) who took part neither in discussions nor further analysis of the results. Careful and systematic analysis, including coding, and interpretative work was conducted by two persons to obtain information at the group level. Conventional content analysis was carried out by two independent coders-OSH experts with more than 20 years of experience with a background in occupational medicine (L.M., I.V.) who were supported by advice from a colleague with a master’s degree in public health (L.P.). 

At first, two of the three researchers (L.M., I.V) in the team read through the data and started to create tentative categories and subcategories. The categories were not built on theoretical considerations but based directly on the data itself. This allowed us to do an unprejudiced assessment and, to our opinion, best fits our research question. Then the same two researchers separately coded the transcript of the first completed focus group (external OSH experts), afterward met to compare their analysis and subsequently to decide on the categories and subcategories which were used for coding of all other focus group transcripts. The subcategories were built up step by step while working through the whole transcription process and were later refined by collapsing and merging initial ones into the final set (see Appendix A Table A5). Then a third independent researcher (L.P.) reviewed all transcripts to verify the findings and to visualize them to be presented as results.

The best supporting text segments (quotes) were captured by all researchers during the coding or reviewing process, afterward discussed and included as anonymous examples to illustrate the different ways responses were expressed. For quotes of employers, the size of the represented company and region is given (Riga, suburbs of Riga represent the biggest city of the country and its surroundings; all others are regions of Latvia); for OSH experts it is specified if the participant is an internal OSH expert or OSH service provider (referred as “external”).

## 3. Results

### 3.1. Quantitative Results

The results from the web-survey show that only 31.5% of surveyed workers mentioned that their employer to some extent had identified existing vulnerable groups and offered to take measures to protect them. This has been one of the least frequent workplace-based preventive measures and was mentioned more than three times less frequently than the most frequent one. Among the most typical preventive measures implemented by the employer provision of additional disinfection and hand-washing materials was mentioned (94.5% of respondents). A total of 79.6% of survey participants reported having distance meetings, 77.5%—having telework in their companies, 76.7%—reorganization of work processes to limit contact with clients, and 76.2%—avoidance of social gathering of workers. If looking for other less frequent measures—control of temperature for workers and visitors and installation of transparent barriers (between workers and clients) should be mentioned (reported by 32.2% and 32.0% of respondents, respectively) (for details on all seventeen analyzed preventive measures see Table A2 given in Appendix A).

When analyzing the answers regarding the protection of vulnerable groups, the results show that only 18.4% of respondents believe that their employer has identified vulnerable groups and offered protection for them in all cases when it was needed. Some more 13.1% have reported this measure partly provided, additional 13.2% respondents–this measure was needed, but not provided. 20.0% of survey participants were not aware if this measure was implemented in the company where they work.

### 3.2. Qualitative Results

Five main themes of information from focus group discussions were identified: (1) acknowledgment of having workers belonging to vulnerable groups, (2) characteristics of workers belonging to vulnerable groups, (3) specific preventive measures implemented to protect vulnerable groups, (4) reasons why employers did not identify vulnerable groups and implement measures, and (5) methods to identify vulnerable groups (see Table A5 in Appendix A).

Although most focus group participants acknowledge having workers belonging to vulnerable groups, many participants mentioned that the companies they represent have not identified any of the vulnerable groups and have not provided any measures to protect these workers. This category of employers is not homogenous—it includes employers who were not aware of the idea of specific protection of vulnerable workers, who did not care about it, who had a high number of such workers as well as who reported a high amount of work, therefore they protected all workers with similar measures:
“*Specific preventive measures for such risk groups are not developed*”- *A large company from Riga, suburbs of Riga*
“*There is a high percentage of workers belonging to this risk group. We have not invented anything special for them*”- *A large company from Riga, suburbs of Riga*
“*Seniors were not highlighted, and those with chronic diseases were also not highlighted*”- *A large company from Vidzeme*
“*We have quite a lot of workers belonging to the risk group. As we had work, we just worked… We did not sort workers*”- *Internal OSH expert*

Discussions on characteristics of workers belonging to vulnerable groups highlighted very different approaches used by employers, however, only one participant described a strategic approach with several groups of workers to be protected:
“*We had three groups—one of those who are older than 60, then—those with chronic respiratory diseases, cardiovascular diseases, cancer patients, and the third one was the group of workers whose family members were health care workers. We identified them as well” *- *A large company from Kurzeme/Zemgale*

Most often focus group participants mentioned elder workers as a typical risk group, however, there is no consensus among the employers and OSH experts what is the borderline age of worker as a critical for belonging to the risk group:
“*Such a specific risk group as workers 65+….*”- *External OSH expert*
“*We look at workers 50+*”- *Internal OSH expert*

Workers with different chronic diseases is another vulnerable group identified by several focus group participants, however, it was interesting that approximately the same number of employers mentioned elder workers and workers with chronic diseases as risk groups, but the number of OSH experts identifying it differed. OSH experts more often specified elder workers than workers with chronic diseases. Few participants mentioned specific health conditions (including respiratory diseases, cardiovascular diseases), others mentioned just having chronic diseases in general or a pregnancy:
“*Those, who have those chronic diseases, frequent upper respiratory infections, heart diseases and so on, yes, they are in the risk group*”- *External OSH expert*
“*In supermarkets … pregnant women, seniors, persons with chronic diseases were invited to apply….*”- *External OSH expert*

Some of the focus group participants mentioned the type of work, the content of the job, etc. that place workers in the risk group. These employers have specifically addressed and protected workers who have a high number of contacts with other persons at work, e.g., client service, passengers, and other workplaces with increased exposure to biological agents:
“*[Workers], who work in client service, …., where every day people are coming and queuing*”- *External OSH expert*
“*To [our] understanding, the risk group consisted of those workers … who have direct contact with passengers. That was the risk group, all the same, what was their age, or health status, we focused on them*”- *A large company from Riga, suburbs of Riga*
“*For us, of course, risk groups are those who clean buses, they are always in the risk group. Also, before COVID they have been exposed to such cases as syringes of addicts and similar cases. They already knew they have risk and must be aware*”- *A large company from Vidzeme*

Some of the participants pointed out that their companies have even expanded the understanding of risk groups by including shift workers (to help them in caring duties for relatives or other persons) and also to family members of workers who have chronic diseases:
“*We specifically addressed shift workers if they have chronic diseases or any other needs…*”- *A large company from Kurzeme/Zemgale*
“*We discussed health status of closest family members—if the workers themselves were not afraid of falling ill, we discussed how we can expose their relatives and friends who are at risk*”- *Small/medium company from Kurzeme/Zemgale*

Several focus group participants mentioned that the risk group for them was different—workers which are critical to their business processes because the business might stop without them causing major problems for the general public at the national level (e.g., no electricity, no heating). Although these were only 5 out of 89 participants, all of them represented companies that are critical for national infrastructure (e.g., supply of electricity, heating):
“*We paid attention to such categories of workers as controllers, … because if they fall ill—we [the city] would be without heating*”- *Internal OSH expert*
“*We were not able to provide telework for our operative workers [controllers, dispatchers]. So, for them, we provided activities to ensure they are healthy*”- *A large company from Riga, suburbs of Riga*
“*In our business in Latvia, the most important thing we had to do was to protect operators, because if the epidemic reaches operators, our business stops*”- *A large company from Kurzeme/Zemgale*

Measures implemented to protect vulnerable groups which were identified during the focus group discussions can be divided into several categories. One of the principles was to ensure that the work of the persons belonging to risk groups is organized away from the direct workplace to avoid contacts with other persons—vulnerable workers were on the priority list for telework (this was the most frequent measure mentioned by the focus group participants) or they were moved away from the areas where contacts with clients were possible but kept working onsite. If none of these mentioned measures was possible, vulnerable workers were offered to use annual paid vacations or paid downtime:
“*… we had workers who told us that they have chronic diseases, …, and asked for telework*”- *A large company from Riga, suburbs of Riga*
“*… part of them chose to use PPEs, part decided to telework, others increased distancing in the workplace*”- *A small/medium company from Latgale*
“*We invited these senior workers to assess their health status and, if possible, work in back-office, meaning, in the place with less or no contacts with clients*”- *A large company from Riga, suburbs of Riga*
“*We tried to move [workers in the risk group] to the warehouse [from the supermarket], in such way decreasing contacts with others*”- *External OSH expert*
“*We talked with those in the risk group and offered to use annual vacations*”- *A small/medium company from Latgale*
“*They [senior workers] were granted unplanned paid holidays. Even food was delivered to their homes”*- *External OSH expert*
“*Already with the first decision of our board we focused on workers 50+, who cannot telework … These workers could have downtime, and it was paid*”- *Internal OSH expert*

Another group of preventive measures pointed out by several focus group participants included awareness-raising activities mainly provided through additional training and discussions with workers on COVID-19 related preventive measures:
“*We very actively communicated with risk groups on the topic “Don’t be a hero!” When you feel any symptoms-…. Stay at home!*”- *A small/medium company from Vidzeme*
“*They were trained on additional preventive measures and that it is extremely important for them to care about themselves*”- *Internal OSH expert*

Several focus group participants reported offering the same preventive measures as provided to other workers, but vulnerable workers were on the priority list to use them. For example, in the situation when at the early stages of the pandemic when there was a shortage of personal protective equipment or disinfectants, vulnerable workers were first to get them. The same principle was applied also when installing transparent plastic barriers:
“*Those workers who are in contacts with passengers, this was a priority for all preventive measures—visors, face masks, disinfectants, we also installed plastic barriers”*- *A large company from Riga, suburbs of Riga*
“*Only in one of [our client] companies, we, to say so, identified special risk groups and provided them with additional personal protective equipment*”- *External OSH expert*

During the focus group discussions of employers another specific measure was identified with the main aim to reduce contacts with possibly infected persons on the way to and from work:
“*Concerning [car] parking, it was strongly advised [to workers] not to use public transportation, parking for private cars paid close to premises”*- *A small/medium company from Riga, suburbs of Riga*

According to the results of the focus group discussions, a comprehensive approach with totally different measures had been implemented for workers critically important for the infrastructure. Their main focus of this preventive approach was to implement measures depending on the level of COVID-19 safety. According to the focus group participants, these companies were ready to exclude all possible contacts of the critical workers with persons outside the workplace during working time and also free time and to provide staying on-site 24/7:
“*We created separate work areas [for critical workers], we provided food, there was a possibility to stay overnight if the situation worsens, but we did not reach such a situation. We reached only the first level of [COVID-19] safety, but we were ready for the next one*”- *A large company from Riga, suburbs of Riga*

Concerning the methods used to identify workers belonging to vulnerable groups, only two focus group participants mentioned the use of data from human resource departments on the age of the workers (through calculations from ID numbers). The most often used method, but reported only by five focus group participants—employers and one external OSH expert was an invitation to apply. Such results from focus group discussions with external OSH experts are surprising as these participants should be able to report experience from several client companies:
“*Ok, we have information on age… But for chronic diseases?*”- *External OSH expert*
“*We specifically invited … workers to apply*”- *A large company from Kurzeme/Zemgale*
“*Pregnant women, seniors, persons with chronic diseases were advised to apply….*”- *External OSH expert*
“*I know from the managers that some of the workers have approached them directly and they have searched individual solutions on how to help the workers. The company should not always know everything, but there must be a way how to find the individual solution*”- *A large company from Kurzeme/Zemgale*
“*We invited those who are close to pre-retirement age or those with any chronic diseases for testing. The company paid for the costs. We invited them if they do not feel safe, to apply and we will search for individual solutions*”- *A small/medium company from Riga, suburbs of Riga*

When discussing the reasons for the low protection of vulnerable workers, only a few focus group participants pointed out specific reasons. The most frequent cause was the poor attitude of workers (e.g., additional personal protective equipment was provided by the employer, but was not used by the workers). Personal data protection issues and lack of information on belonging to risk groups were mentioned by some participants:
“*It seems to me that the most difficult was to change existing customs. The way the workers are used to doing things and changing focus…*”- *A large company from Kurzeme/Zemgale*
“*We offered additional personal protective equipment–but senior workers quickly refused to use them, they said—it is easier to work without them*”- *External OSH expert*
“*We deeply respected that persons are different, that risks for persons and their family members are different, but we do not know all of these situations*”- *A large company from Kurzeme/Zemgale*
“*We understand that this information [on the chronic diseases] is confidential*”- *A large company from Riga, suburbs of Riga*
“*On chronic diseases—such …. precise information is not available for the employer*”- *External OSH expert*
“*They are in the risk group, but…. This information is confidential and nobody shares it*”- *External OSH expert*

## 4. Discussion

Although already at the very beginning of the COVID-19 pandemic, it was clear that elderly persons and persons with chronic diseases have elevated risk for severe progress of COVID-19 infection and other poor outcomes, quantitative and qualitative data of our study show that identification and protection of these persons at workplaces were not sufficient [2]. According to our opinion, the most worrying data are those that show that 13.2% of respondents report that this measure was needed, but not provided, additionally, 20.0% were not aware if this measure was implemented. We believe that the group of respondents who reported that this measure is not needed is not homogenous—it includes respondents from companies that do not employ workers belonging to the vulnerable group and those who do not acknowledge that vulnerable workers need specific attention. In general, such results support earlier studies—there is a huge possibility and an urgent need for workplace interventions to cover the protection of vulnerable workers against COVID-19 related risks [8,16].

Another worrying aspect is the fact that the understanding of the term “vulnerable workers” by focus group participants is very heterogeneous, and this group includes workers of different ages (but older than 50, workers with very different diseases, pregnant women, shift workers, workers critical for infrastructure companies, workers having health care workers in their households, workers in client service, etc.). Therefore, all of the measures identified within the focus group discussions do not specifically apply to elderly persons and persons with chronic diseases which have been reported as persons with elevated risk for severe progress of COVID-19 infection and other poor outcomes [2]. To our understanding, COVID-19 has highlighted the need to move away from general population-based workplace risk assessment to an individually personalized approach covering both—individual aspects of workers (including health based) and type of work involving increased contacts with COVID-19 positive persons [8].

When looking at the reasons for such low protection of vulnerable workers, this study shows that both employers and OSH experts do not recognize the importance of taking care of vulnerable workers which might be linked to the lack of knowledge and understanding of possible methods for doing workplace risk assessment in emergencies. In such circumstances, advice from the governments and non-governmental organizations needs to be carefully considered in managing occupational risks from COVID-19 [8].

It seems that another reason for low protection is low interest in looking for information that might help to identify vulnerable workers, e.g., the use of data available to human resource departments on the age of the workers. Such results seem to be paradoxical as all of the employers have access to the data on the age of the worker as according to the national legislation in Latvia ID number of most persons contains birth date.

In addition, simple communication between the employer and workers like reasonably explained invitations to apply if the worker would like to be protected, but without asking for the underlying reasons was mentioned only by five focus group participants. This seems to be especially surprising if looking at the results from focus group discussions with external OSH experts as these participants should be able to report experience from several client companies. This raises questions on the style of communication on OSH topics within companies, with OSH experts as well as the quality of the provided external OSH services (e.g., if they do cover health-related aspects of OSH in sufficient detail).

There have been publications that stress the central role of occupational physicians in the individual approach for the protection of vulnerable workers [8]. Although we acknowledge the possible role of occupational physicians, the national legal system of medical surveillance of workers plays an important role. In typical circumstances, the OSH system at the company level in Latvia is fully separated from the medical surveillance of workers as this surveillance is provided by the health care system. The link between the workplace and health care is a document providing information on existing workplace risk factors (including biological risks) at the particular workplaces to the occupational physician who provides risk factor-specific medical examinations and reports back to the employer if the worker is fit for the particular job or not. During the first COVID-19 emergency state all out-patient health care establishments were closed by the government decision and health surveillance for workers was not provided [26]. Therefore, the current legal system for medical surveillance of workers is not suitable for the involvement of occupational physicians in the identification of vulnerable workers as well as the selection of appropriate preventive measures. Further, there is a potential problem that external OSH experts are not sufficiently addressing heath related risks, e.g., biological risks.

We found that most often measures provided by the employers for the vulnerable workers can be grouped under the umbrella—taking vulnerable workers away from the workplace where direct contact with other persons (including co-workers) is possible. This includes telework, use of annual paid holidays, and paid downtime (in total mentioned 33 times during the focus group discussions). Such measures seem very logical, however, not all of them are sustainable in the long-term from the business perspective. For example, annual paid holidays are fine to be used one or two months, but are not applicable when the emergency state repeats and lasts for months. The same is applicable for the downtime paid by the employer—from one side these employers most probably will not be able to pay it in long-term, from the other side it can cause internal dissatisfaction between the workers (in order to have equal income some of the workers will have to work, but others (the vulnerable ones) will be allowed to stay at home). In such a context the employers should look for the possibilities to requalify vulnerable workers that they can do some work and earn their salary.

Only a few focus group participants mentioned that they provided vulnerable workers with additional personal protective equipment. These low results most probably can be explained by the fact that the employers did not specifically focus on personal protective equipment for vulnerable workers, but treated all workers equally. Another part of the focus group discussions which is not the focus of this article covered topics related to the use of personal protective equipment. It allowed us to identify the following problems: the shortage of personal protective equipment in general in the very early stages of the COVID-19 pandemic, availability of personal protective equipment due to the priority of supply to health care, quality of the personal protective equipment available on the market, the rise of costs of all types of personal protective equipment were the main problems described by the employers and OSH experts [26]. Similar aspects were also risen in mass media; however, their main focus was on low protection of health care workers [26].

While analyzing the results from the focus group discussions, we found that employers provide more preventive measures to identify and protect their critical workers based on their business functions and not on the employment level in the company (e.g., belonging to the management). Companies, where such workers exist, were ready to provide a full board on-site for maintenance of their businesses (like the supply of electricity and heating). If we would have excluded these answers from the analysis, the average protection level of vulnerable workers might have been even lower, however, we were not able to extract from the analysis all the answers from these companies because they have protected their other vulnerable workers as well.

When looking at the results of our study in terms of limitations, we have identified several of them. One of the limitations of the web-survey method is the fact that some groups of workers may be excluded from the sample by default (e.g., elderly, people living in remote areas, and people with low education and digital literacy) [21]. In addition, our questionnaire was available only in Latvian, and it might have caused less response rate from the side of the Russian-speaking population. A non-probability sampling method we used to gather survey data is another limitation of the study. The advantage of this method is the possibility to quickly gather information from respondents which was important because of the implementation requirements of the project “Life with COVID-19: Evaluation of Overcoming the Coronavirus Crisis in Latvia and Recommendations for Societal Resilience in the Future” [21]. To overcome this limitation at least partly and to obtain data that is representative of the demographic profile of the working population in Latvia, the sample was weighted based on gender and age. We were not able to weight data in terms of education or work experience as such population estimates were not available from the Central Statistical Bureau of Latvia for the study period. Despite these limitations, the survey provides descriptive information and useful insights into the protection of vulnerable groups of workers during the COVID-19 pandemic.

Regarding focus group discussions the most challenging limitation was the fact that we struggled to recruit representatives from companies with less known brands or companies who have not previously shared their experience in OSH. Our participants were recruited among OSH experts already previously active in the OSH community or representing companies willing to share their experience and good practice in OSH. This might mean that focus group participants represented the companies with OSH performance level which was significantly above the average already before COVID-19 and the actual situation in other companies might be even worse.

During the interviewing period legal restrictions regarding gathering of persons from different households changed, therefore, mixed focus group discussions were carried out: part of the participants participated onsite, part—online (for details see Table A3 and Table A4 in Appendix A). Although experienced interviewers tried to ensure equal opportunities for all participants, e.g., by addressing questions to persons individually and trying to engage them, this might have influenced the equality of every single participant in all discussed questions.

## 5. Conclusions

COVID-19 pandemic has shown that neither employers nor OSH experts were ready for the management of new and emerging risks at the workplace level. This allows us to raise the question if the traditional OSH management systems and approaches where safety engineers play a crucial role will be ready for the management of other unpredictable risks or the protection of specific vulnerable groups in future, especially with regards to biological risks. In addition, the role of occupational physicians as part of occupational health services should be strengthened, especially when the protection of workers with chronic diseases is required.

There is an urgent need for adequate health and safety arrangements and provision of protection for essential workers. Most employers do not see elderly workers and workers with chronic diseases as risk groups, thus they are not implementing measures to protect them. This might be an explanation why the second wave of COVID-19 hit Latvia more severely than the first wave, including several outbreaks of COVID-19 in companies. Poor communication and lack of interest of employers and OSH experts to ask their workers if they need special protection is the topic to be addressed not only at the company and national level, but probably also at the international level. It should include the promotion of personalized workplace risk assessment covering both—individual aspects of workers and type of work involving increased contacts with COVID-19 positive persons as well as appropriate health behaviors.

## Data Availability

Data set from the web-survey is available upon request to the corresponding author (L.M.). Data sharing of focus group discussions is not applicable as the data consists of focus group transcripts, which for reasons of confidentiality, cannot be shared.

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
