# Peer review of "Reasons for Low Protection of Vulnerable Workers from COVID-19—Results from the Quantitative and Qualitative Study on Working Life in Latvia"

_ijerph, 2021, doi:10.3390/ijerph18105188_

Round 1

Reviewer 1 Report

The aim has been to identify preventive measures provided by the employers for the vulnerable groups during the emergency situation caused by COVID-19. In addition, some of the reasons why no additional measures were used have been discussed.

The objective is interesting and the use of qualitative and quantitative methodology is of great value in order to identify specific needs and difficulties.

I think that the information obtained is very useful, not only for the current situation, but also for possible similar future situations.However, I would like the authors to provide more information in the methodology section.

-It would be helpful for the reader to include a specific section on participants, where the samples of each part of the work are described independently. It would also be helpful if the inclusion and exclusion criteria for each of the samples were clarified.

- More information is needed on how the sample was selected, for example, was a sample size calculation made for the quantitative study? After reviewing the sample, could we say that there is adequate representation of the working population of Latvia? For example, in the case of the sample of the quantitative study, there is a high participation of women with a high cultural level. Could this sample not be representative of the country's workers?

- With respect to the survey, how did you build the statements included in the survey? What was the basis for including these statements? Previous studies? Measures developed in the country or in other countries?

In summary, in my opinion, it is an interesting study that has been carried out with an adequate methodology and that provides useful information.

Reviewer 2 Report

Thank you for the opportunity to review the manuscript “Reasons for low protection of vulnerable workers from COVID-19 - results from the study on working life in Latvia”.

Congratulations to the authors for their work, I found your paper a potentially very valuable resource on Health Science and therefore an interesting and relevant contribution to IJERPH.

The manuscript identifies measures taken by the employer to protect them and to investigate reasons for low protection during the 1st wave of the COVID-19 pandemic.

However, in my opinion there are several aspects should be revised as noted below.

GENERAL COMMENTS:

Extensive editing of English language and style required.

SPECIFIC COMMENTS:

TITTLE

Include study design in title.

ABSTRACT

In the abstract the results and conclusions seem mixed. It is recommended to structure the information.

INTRODUCTION

Correct.

METHOD, RESULTS AND DISCUSSION

  • In the quantitative study, the authors must register their sample universe and put the response rate obtained. A flow chart can help you understand the loss of participants. It's funny that they are just 1000.
  • The work has a great methodological weakness, the creation and validation of the measurement instrument used in the quantitative study. Please explain how the instrument was created. Is it reliable and valid?
  • Please put the evaluation instrument as supplementary material. It is important to know the items used.
  • In the text remove the reference to M+-SD as it is not correct, it should be M (SD) (Page 4 line 180).
  • In the verbatim of the participants, is anonymity maintained?
  • In the discussion section, please add the limitations of the study (there are several limitations such as sampling, etc.).

CONCLUSION

It mixes with the discussion. Put the conclusions of the study. Do not compare with other studies, that's what the discussion is for.

Reviewer 3 Report

I have carefully read the paper entitled " Reasons for low protection of vulnerable workers from COVID-19 - results from the study on working life in Latvia" .

The paper is well written overall but needs some changes to make it publishable.

  1. The reasons for the low use of PPE are clear. Considering that we do not know the local situation, it would be necessary to know what the lack of information from employers is due to. Did the media give prominence to the pandemic data?
  2. Has there been a shortage of PPE as in other countries?
  3. At least in the introduction, greater emphasis should be given to doctors. Has the situation regarding health professionals been homogeneous or some categories have been more affected by the pandemic (see and introduce a parallelism with Nioi, Matteo, et al. "COVID-19 and Italian healthcare workers from the initial sacrifice to the mRNA vaccine: Pandemic chrono-history, epidemiological data, Ethical Dilemmas, and Future Challenges." Frontiers in Public Health 8 (2020).]
  4. The paper often speaks of "legal obligations". Some considerations are required to be included in the introduction.
  5. a) Lithuania has introduced a legal shield for doctors? (cf.Aloja, Ernesto, et al. "COVID-19 and medical liability: Italy denies the shield to its heroes." EClinicalMedicine 25 (2020).)
  6. b) Regarding exposed workers, does your state provide compensation / compensation? And what obligations are / were there with regard to the use of PPE?
  7. E) Probably the exposure of the reasons given by the employers weighs down the part of the results. I also suggest a synthesis through figures / tables.

Round 2

Reviewer 3 Report

I carefully read the work entitled  “Reasons for low protection of vulnerable workers from COVID-19 - results from the quantitative and qualitative study on working life in Latvia”. Unfortunately, most of the requests made in the previous round were not made. Unfortunately, the number of works on COVID-19 is starting to be very high and a high quality is needed for the works to be considered for publication. Although I appreciated the changes applied, I do not think the paper is still publishable.

For clarity, the changes should be addressed as follows:

Major concerns

  1. The bibliography and the premises given in the introduction are not sufficient. Review the text with the indications provided in the previous round.
  2. The paper focuses on low protection of vulnerable workers from COVID-19. The authors should make an exemplary parallelism with a foreign reality / country where this category has been better protected.
  3. In the "Research Manuscript Sections" section of Instructions to Authors it literally says: "Provide a concise and precise description of the experimental results, their interpretation as well as the experimental conclusions that can be drawn." Do the authors believe that the description of the results of their paper is precise and concise? The description of "qualitative results" (chapter 3.2 ) is written in a journalistic style and this form is unacceptable in a scientific journal. The authors reported the individual sentences attributable to the different employers (i.e. “…We invited these senior workers to assess their health status and, if possible, work in back-office,meaning, in the place with less or no contacts with clients” - A large companies from Riga, suburbs of Riga ….We tried to move [workers in the risk group] to the warehouse [from the supermarket], in such way decreasing contacts with others”  - External OSH expert  “We talked with those in the risk group and offered to use annual vacations”  - A small/medium company from Latgale “They [senior workers] were granted unplanned paid holidays. Even food was delivered to their  homes”).

I consider the presentation of the results to be a fundamental section of the work. The authors were asked to change it in the previous round as well. The persistence of this grotesque description does not make the article suitable for publication in a scientific journal.

Minor concerns

Line 375 The table with no data in the center of the text must be corrected.

I do not consider the paper in its current form to be considered for publication but I am confident that the authors will take advantage of this last chance and make the required changes in a short time.

Author Response

Thank you for reviewing our manuscript “Reasons for low protection of vulnerable workers from COVID-19 – quantitative and qualitative results from the study on working life in Latvia”, we have prepared answers to your comments as no major modifications were made in the text of the article (except for the deleting the table in Line 375).
We have already in the previous round explained the reasons for not covering aspects of protection of health care workers in the bibliography, which are mainly related to the focus of the article on vulnerable workers (elder workers, workers with chronic diseases) and not on the front-line workers (medical professionals). Also, the situation during the COVID-19 pandemic was principally different in health care institutions in Latvia where the full range of preventive measures was encouraged and ensured by government and employers covering also any worker belonging to vulnerable groups. So the inclusion of this sector and the situation regarding frontline workers in this article would give the research community wrong information and conclusions as they were radically different. 
While drafting the article, we did a thorough literature review to identify research done in the context of protection of the vulnerable workers during the COVID-19, however, due to the novelty of the problem, only several theoretical articles were identified and they were used for the theoretical background. No published data were identified on practical measures taken by the employer at the company level (therefore, making parallelism was impossible). Examples of identified articles included as references in our article are as follows:
1.    Baker, M.G.; Peckham, T.K.; Seixas, N.S. Estimating the Burden of United States Workers Exposed to Infection or Disease: A Key Factor in Containing Risk of COVID-19 Infection. PLoS ONE 2020, 15, e0232452. [Google Scholar] [CrossRef] [PubMed]
2.    Larochelle, M.R. “Is It Safe for Me to Go to Work?” Risk Stratification for Workers during the Covid-19 Pandemic. N. Engl. J. Med. 2020, 383. [Google Scholar] [CrossRef] [PubMed]
3.    Coggon, D.; Croft, P.; Cullinan, P.; Williams, A. Assessment of Workers Personal Vulnerability to Covid-19 Using Covid-Age. Occup. Med. 2020, 70, 461–464. [Google Scholar] [CrossRef] [PubMed]
Regarding the description of the results of our paper, we would like to mention that before submitting our article to the Int. J. Environ. Res. Public Health, we thoroughly reviewed if this journal publishes articles based on focus group discussions and how the results are presented in these cases. As examples on occupational health and safety or environmental safety, we would like to mention the following articles:
1.    Pyo, J.; Lee, M.; Ock, M.; Park, G.; Yang, D.; Park, J.; Kim, Y. Bus Workers’ Experiences with and Perceptions of a Health Promotion Program: A Qualitative Study Using a Focus Group Discussion. Int. J. Environ. Res. Public Health 2020, 17, 1992. https://doi.org/10.3390/ijerph17061992
2.    Uhl, M.; Santos, R.R.; Costa, J.; Santos, O.; Virgolino, A.; Evans, D.S.; Murray, C.; Mulcahy, M.; Ubong, D.; Sepai, O.; Lobo Vicente, J.; Leitner, M.; Benda-Kahri, S.; Zanini-Freitag, D. Chemical Exposure: European Citizens’ Perspectives, Trust, and Concerns on Human Biomonitoring Initiatives, Information Needs, and Scientific Results. Int. J. Environ. Res. Public Health 2021, 18, 1532. https://doi.org/10.3390/ijerph18041532 
Based on our best knowledge, previous experience, and the analyses of the articles where focus group discussions have been used as a research method, we decided on the approach to the research questions, structure, and style of the article. We have also chosen to present the problem using both quantitative and qualitative methods to provide wider insight into our results. We regret that you believe that the article is not suitable for publication in a scientific journal.
Thanks once more for investing your time and effort to improve our scientific output!